

# Effects of L-carnitine supplementation for women with polycystic ovary syndrome: a systematic review and meta-analysis

Mohd Falihin Mohd Shukri, Mohd Noor Norhayati, Salziyan Badrin and Azidah Abdul Kadir

Department of Family Medicine, Universiti Sains Malaysia, Kubang Kerian, Kelantan, Malaysia

## ABSTRACT

**Background**. Polycystic ovary syndrome (PCOS) is a disorder in reproductive age women and is characterized by hyperandrogenic anovulation and oligo-amenorrhea, which leads to infertility. Anovulation in PCOS is associated with low follicle-stimulating hormone levels and the arrest of antral follicle development in the final stages of maturation. L-carnitine (LC) plays a role in fatty acid metabolism, which is found to be lacking in PCOS patients. This systematic review and meta-analysis aimed to determine the effectiveness of LC supplementation for patients with PCOS.

**Methods**. We searched the Cochrane Central Register of Controlled Trials (CENTRAL), MEDLINE, Embase, Cumulative Index to Nursing and Allied Health Literature (CINAHL), Psychological Information Database (PsycINFO), and the World Health Organization International Clinical Trials Registry Platform for all randomized control trials, comparing LC alone or in combination with other standard treatments for the treatment of PCOS from inception till June 2021. We independently screened titles and abstracts to identify available trials, and complete texts of the trials were checked for eligibility. Data on the methods, interventions, outcomes, and risk of bias from the included trials were independently extracted by the authors. The estimation of risk ratios and mean differences with a 95 percent confidence interval (CI) was performed using a random-effects model.

**Results**. Nine studies with 995 participants were included in this review. Five comparison groups were involved. In one comparison group, LC reduced the fasting plasma glucose (FPG) (mean differences (MD) $-5.10$, 95% CI [$-6.25$ to $-3.95$]; $P = 0.00001$), serum low-density lipoprotein (LDL) (MD $-25.00$, 95% CI [$-27.93$ to $-22.07$]; $P = 0.00001$), serum total cholesterol (MD $-21.00$, 95% CI [$-24.14$ to $-17.86$]; $P = 0.00001$), and serum triglyceride (TG) (MD $-9.00$, 95% CI [$-11.46$ to $-6.54$]; $P = 0.00001$) with moderate certainty of evidence. Another comparison group demonstrated that LC lowers the LDL (MD $-12.00$, 95% CI [$-15.80$ to $-8.20$]; $P = 0.00001$), serum total cholesterol (MD $-24.00$, 95% CI [$-27.61$ to $-20.39$]; $P = 0.00001$), and serum TG (MD $-19.00$, 95% CI [$-22.79$ to $-15.21$]; $P = 0.00001$) with moderate certainty of evidence.

**Conclusion**. There was low to moderate certainty of evidence that LC improves Body Mass Index (BMI) and serum LDL, TG, and total cholesterol levels in women with PCOS.

Corresponding author
Salziyan Badrin, salziyan@usm.my

## INTRODUCTION

Polycystic ovary syndrome (PCOS) is a common disease that affects women of reproductive age with a prevalence ranging between 6.5 and 8 percent (*Norman et al., 2007*). It is an endocrine disorder that presents with irregular menses, hyperandrogenism, and polycystic ovaries. The clinical presentation includes oligomenorrhea or amenorrhea, hirsutism, and infertility (*Sirmans & Pate, 2013*). Anovulatory PCOS is associated with low follicle-stimulating hormone levels and the arrest of antral follicle development in the final stages of maturation (*Badawy & Elnashar, 2011*). The diagnosis of PCOS is based on the criteria defined by the Rotterdam European Society for Human Reproduction (ESHRE) and American Society of Reproductive Medicine (ASRM), which is currently known as the Rotterdam Criteria. The criteria comprise three features, including oligo or amenorrhea, clinical and biochemical signs of hyperandrogenism, and evidence of polycystic ovaries on ultrasound findings. Two out of three features confirm the diagnosis of PCOS (*Badawy & Elnashar, 2011*). Polycystic features of the ovary on ultrasound suggest PCOS when 12 or more follicles in each ovary measure 2–9 mm in diameter and/or increased ovarian volume (*Badawy & Elnashar, 2011*). Obesity is highly prevalent in PCOS women, and it is an independent risk factor for coronary artery disease, as obesity is associated with insulin resistance, dyslipidemia, and ovulatory dysfunction in adolescents (*Traub, 2011*). The evaluation of risk factors for coronary arterial diseases (CADs) is essential in PCOS, as CADs entail the greatest long-term risk for PCOS (*Traub, 2011*).

Medications such as clomiphene citrate, tamoxifen, aromatase inhibitors, metformin, glucocorticoids, gonadotropins, or laparoscopic ovarian drilling can be used to alleviate the anovulation problem faced by PCOS patients (*Badawy & Elnashar, 2011*) (5). L-carnitine (LC) is an endogenous compound synthesized by the human body, and it plays a key role in fatty acid metabolism (*Johri et al., 2014*). Carnitine is synthesized from lysine and methionine and is available from dietary sources such as meat, poultry, and dairy products (*Johri et al., 2014*). Carnitine acts as an obligatory cofactor for the oxidation of fatty acids by facilitating the transportation of long-chain fatty acids across the mitochondrial membrane. LC level is low in patients with PCOS; therefore, the use of LC as an adjunctive therapy in the management of insulin resistance or obesity in women may be beneficial (*Celik et al., 2017*). LC can boost ovarian function and decrease oxidative stress and inflammation. Furthermore, LC can normalize androgen levels, contributing to a significant decrease in testosterone levels (*Della Corte et al., 2020*). LC may enhance insulin sensitivity, thereby affecting the levels of androgens and ovarian hormones (*Maleki et al., 2019*).

This systematic review and meta-analysis aimed to determine the effectiveness of LC supplementation for patients with PCOS. The primary outcomes were clinical pregnancy and ovulation rate, Body Mass Index (BMI), fasting plasma glucose (FPG), and serum lipid levels, including low-density lipoprotein (LDL), triglycerides (TGs), total cholesterol, and high-density lipoprotein (HDL) levels. Mental health status, serum follicular stimulating hormone (FSH), and luteinizing hormone (LH) levels comprised the secondary outcomes. This review could reveal evidence of alternate therapy for improving clinical pregnancy outcomes and metabolic indicators in PCOS patients.

The effects of LC supplementation information may aid physicians in selecting and deciding on an alternate supplement to enhance PCOS metabolic indicators and increase clinical pregnancy rates.

## MATERIAL AND METHODS

The methodology and reporting conducted in this review are based on the guidelines recommended by the Cochrane Collaboration in the Cochrane Handbook for Systematic Reviews of Interventions (*Higgins et al., 2021*). The quality of evidence was evaluated according to the Grading of Recommendation Assessment, Development and Evaluation (GRADE) guidelines (*Guyatt et al., 2008*).

### Identification and eligibility of study

All randomized control trials (RCTs) comparing LC alone or in combination with other standard medications or other dietary supplements for the treatment and supplementation of PCOS women were considered in the review. The comparators were selected according to the availability of comparative studies versus LC. The participants included women who had been diagnosed with PCOS based on the revised ESHRE and the ASRM diagnosis of PCOS, according to the Rotterdam criteria of 2003. We excluded cross-over trials and studies other than RCTs. Werestricted the publications to the English language only.

We used the search strategy in Appendix S1 and searched through Cochrane Central Register of Controlled Trials (CENTRAL), MEDLINE, Embase, Cumulative Index to Nursing and Allied Health Literature (CINAHL), Psychological Information Database (PsycINFO), and the World Health Organization International Clinical Trials Registry Platform for all available studies comparing LC alone or in combination with other standard treatments to treat PCOS. For additional datasets, we modified the search strategy. Using the Boolean operators AND as well as OR, we combined the terms "polycystic ovarian syndrome" and "L carnitine" (refer to Appendix S1). To locate unpublished trials or trials that could not be found using electronic searches, we looked through the reference lists of recognized RCTs and read the relevant articles. We also reached out to experts in the field and used the World Health Organization International Clinical Trials Registry Platform (http://www.who.int/ictrp/en/) and http://www.clinicaltrials.gov to find active trials.

Three authors (MFMS, SB, AAK) scanned the repository of articles for trial selection from the titles and abstracts derived from the searches. Therein, we obtained full-text articles when they appeared to meet the eligibility criteria or when there was insufficient information to assess the eligibility. We documented the reasons behind exclusion after the authors independently reviewed the eligibility of the studies. Any differences were settled by discussion among the authors. If more information is required, then we will contact the authors. We utilized the procedure recommended by the Cochrane Handbook for Systematic Reviews of Interventions for searching and selecting studies (*Higgins et al., 2021*).

We retrieved 56 records from the search of the electronic databases, 22 records from Cochrane, 30 from MEDLINE, and four records from other databases. We screened 33 records after removing duplicates. Furthermore, we reviewed the complete text of 28

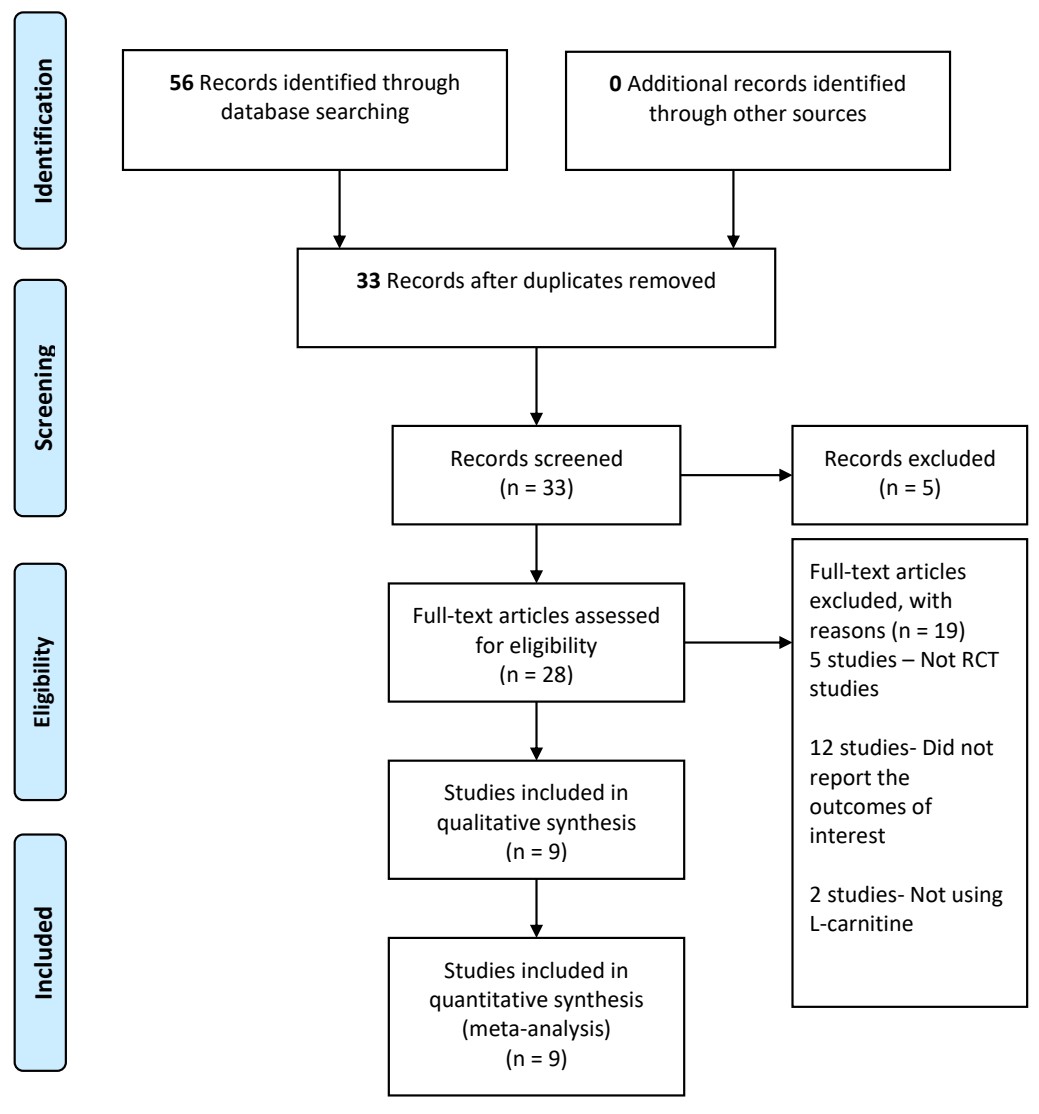

**Figure 1 PRISMA study flow diagram.**

records—nine studies met the inclusion criteria, whereas 19 studies did not fulfill the inclusion criteria and were, therefore, excluded (refer to Fig. 1). The number of records retrieved, screened, included, and excluded was presented in the PRISMA study flow diagram (Fig. 1).

## Data collection and analysis

Three authors independently extracted data. We extracted data on the study setting, participant characteristics (age), methodology (inclusion and exclusion criteria, number of participants randomized and analyzed, and duration of follow-up), description of interventions used (dose, frequency, preparation, and duration used), and the measured outcomes. We also extracted data pertaining to the number of intrauterine gestational

sacs and fetal heart rate visible by transvaginal ultrasound within 12 weeks of intervention (clinical pregnancy rate), the number of visible leading follicles of more than or equal to 18 mm by transvaginal ultrasound within 12 weeks of intervention (ovulation rate), BMI in kg/m2, serum LDL, serum HDL, TG, total cholesterol in mmol/l or mg/dl, and fasting blood glucose (FPG) in mg/dl, serum FSH and LH in IU/L, mental health status assessment using any questionnaires, and adverse side effects such as gastrointestinal disturbances (abdominal pain, nausea, and vomiting. Disagreements between the review authors (MFMS, SB, AAK) were resolved by discussion with the fourth author (NMN).

## Assessment risk of bias

We used the Cochrane Collaboration's risk-of-bias tools to assess the risk of bias in the included studies (*Higgins et al., 2021*). Three authors (MFMS, SB, AAK) assessed the selection bias (randomization and allocation concealment), performance bias (blinding of participant and health personnel), detection bias (blinding of outcome assessment), attrition bias (incomplete outcome data), reporting bias (selective reporting), and other biases (recall bias, transfer bias and etc.) independently. We classified the risk of bias as very low, low, moderate, or high. We also resolved disagreements by conducting discussions with the fourth author (NMN). In addition, we assessed the quality of evidence for primary and secondary outcomes, according to the GRADE methodology for risk of bias, inconsistency, indirectness, imprecision, and publication bias and classified it as very low, low, moderate, or high (*Guyatt et al., 2008*). Furthermore, we assessed the presence of the risk of bias, inconsistency or unexplained heterogeneity, indirectness of evidence, imprecision, and publication bias. We classified them as very low, low, moderate, and high.

## Statistical analysis

We analyzed data using Review Manager 5.4 software (*Manager, 2020*) for the statistical analyses. Moreover, we used a random-effects model to pool data. We measured the treatment effect using risk ratios (RR) for dichotomous outcomes and mean differences (MD) with 95 percent confidence intervals (CI) for continuous outcomes. We assessed the presence of heterogeneity in two steps. First, the assessment was performed at face value by comparing populations, settings, interventions, and outcomes (*Higgins et al., 2021*). Second, the statistical heterogeneity was assessed by using the $I^2$ statistic (*Higgins et al., 2021*). We used the interpretation of heterogeneity as follows: 0% to 40% might not be important; 30% to 60% may represent moderate heterogeneity; 50% to 90% may represent substantial heterogeneity; and 75 percent to 100 percent would indicate considerable heterogeneity (*Higgins et al., 2021*). We checked the included trials for the unit of analysis errors. The unit of analysis errors can occur when trials randomize participants to intervention or control groups in clusters but analyze the results using the total number of individual participants. Based on the mean cluster size and intra-cluster correlation coefficient, we adjusted the results from trials with the unit of analysis errors (*Higgins et al., 2021*). Thereafter, we contacted the trial's original authors to request data that had been missing or incorrectly reported. If missing data was not accessible, we conducted analyses using the available data. We performed a sensitivity analysis to investigate the impact of the

high risk of bias on sequence generation and allocation concealment of included studies. If there were sufficient studies, then we used funnel plots to assess the possibility of reporting biases or small study biases, or both.

GRADEPro software was used to analyze the quality of evidence or certainty in the body of evidence for each outcome, and we classified the quality of evidence as high, moderate, low, and very low.

# RESULTS

## Trial selection

We retrieved 56 records from the electronic searches that were available from inception until June 2021. We screened a total of 33 records after duplicates were removed, and we excluded five studies that did not meet the eligibility criteria. Out of these 28 studies, another 19 studies were excluded. Five out of 19 studies were not RCT studies (*Celik et al., 2017*; *Eyupoglu et al., 2019*; *Fenkci et al., 2008*; *Maleki et al., 2019*; *Salehpour et al., 2019*), and 12 studies were excluded because they did not report outcomes of interest for this review (*Chen et al., 2020*; *Chen et al., 2016*; *Cree-Green et al., 2019*; *Dong et al., 2015*; *Hamed, 2016*; *Jia et al., 2019*; *Karakas et al., 2016*; *Selen Alpergin et al., 2017*; *Sheida et al., 2021*; *Sun et al., 2019*; *Vonica et al., 2019*; *Zhao et al., 2015*). Two other studies reported the effects of other supplementations other than LC and did not fulfil the eligibility criteria (*Nct, 2019*; *Vigerust et al., 2012*). We have summarized the results of the search strategy in Fig. 1.

## Characteristics of included trials

We included nine trials with a total of 995 participants (*El Sharkwy & Sharaf El-Din, 2019*; *El Sharkwy & Abd El Aziz, 2019*; *Ismail et al., 2014*; *Jamilian et al., 2017*; *Jamilian et al., 2019a*; *Jamilian et al., 2019b*; *Kortam, Abdelrahman & Fateen, 2020*; *Samimi et al., 2016*; *Talari et al., 2019*). All nine trials recruited women who had been diagnosed with PCOS based on the Rotterdam criteria. Six trials involved the participants aged between 18 and 40 years (*El Sharkwy & Abd El Aziz, 2019*; *Jamilian et al., 2017*; *Jamilian et al., 2019a*; *Jamilian et al., 2019b*; *Samimi et al., 2016*; *Talari et al., 2019*). On the other hand, two trials include BMI >25 kg/m2 as one of the inclusion criteria (*Jamilian et al., 2019b*; *Samimi et al., 2016*), and three trials used clomiphene citrate resistant PCOS as the inclusion criteria (*El Sharkwy & Sharaf El-Din, 2019*; *El Sharkwy & Abd El Aziz, 2019*; *Ismail et al., 2014*). All nine trials reported hyperprolactinemia as the exclusion criteria. Eight trials excluded participants with endocrine disorder, and the duration of the study was 12 weeks, with the exception of one trial (*Kortam, Abdelrahman & Fateen, 2020*) that did not mention the study duration. Four out of nine included trials excluded women who were pregnant in the trial (*Jamilian et al., 2017*; *Jamilian et al., 2019a*; *Jamilian et al., 2019b*; *Talari et al., 2019*). Three studies excluded diabetic patients as participants in the trial (*Jamilian et al., 2019a*; *Jamilian et al., 2019b*; *Samimi et al., 2016*).

## Outcomes

The nine included trials had diverse groups, which addressed various comparisons and outcomes, resulting in several comparisons that contributed to each of predefined

outcomes. All studies had methodological limitations, and there were too few studies to allow pooling of all primary and secondary outcomes.

Four included trials reported on the clinical pregnancy rate and the ovulation rate (*El Sharkwy & Sharaf El-Din, 2019*; *El Sharkwy & Abd El Aziz, 2019*; *Ismail et al., 2014*; *Kortam, Abdelrahman & Fateen, 2020*), whereas seven out of nine included trials reported BMI (*El Sharkwy & Sharaf El-Din, 2019*; *El Sharkwy & Abd El Aziz, 2019*; *Jamilian et al., 2017*; *Jamilian et al., 2019b*; *Kortam, Abdelrahman & Fateen, 2020*; *Samimi et al., 2016*; *Talari et al., 2019*). The lipid profile, including serum LDL, HDL, total cholesterol, and TG levels, were reported in four trials (*El Sharkwy & Sharaf El-Din, 2019*; *El Sharkwy & Abd El Aziz, 2019*; *Jamilian et al., 2019b*; *Samimi et al., 2016*), and FPG was reported in four trials (*El Sharkwy & Sharaf El-Din, 2019*; *El Sharkwy & Abd El Aziz, 2019*; *Jamilian et al., 2019b*; *Samimi et al., 2016*).

Five trials reported secondary outcomes, which are hormonal levels, including the serum FSH levels, and LH levels, and mental health status. The serum FSH and LH levels were reported in three trials (*El Sharkwy & Sharaf El-Din, 2019*; *El Sharkwy & Abd El Aziz, 2019*; *Kortam, Abdelrahman & Fateen, 2020*), and the mental health status was reported in two trials (*Jamilian et al., 2017*; *Jamilian et al., 2019a*).

## Assessment risk of bias

The assessment of risk of bias has been presented in Figs. 2 and 3. The details of these trials are summarized in Table 1. All nine trials described the method of randomization used. Eight trials randomized the participants using computer-generated randomization (*El Sharkwy & Sharaf El-Din, 2019*; *El Sharkwy & Abd El Aziz, 2019*; *Ismail et al., 2014*; *Jamilian et al., 2017*; *Jamilian et al., 2019a*; *Jamilian et al., 2019b*; *Samimi et al., 2016*; *Talari et al., 2019*), with the exception of one trial (*Jamilian et al., 2019b*) in which the randomization sequence was manually executed at the clinic. Therefore, we judged a high risk of random sequence generation bias for this trial (*Jamilian et al., 2019b*), whereas a low risk of bias was assigned to the other eight trials. Allocation concealment was reported in all trials. All trials conducting the study using placebo capsules, which were designed to be identical to LC capsules. Three trials (*El Sharkwy & Sharaf El-Din, 2019*; *El Sharkwy & Abd El Aziz, 2019*; *Ismail et al., 2014*) distributed the capsules using opaque and sealed envelopes. Therefore, for allocation concealment, all trials had a low risk of bias. Eight trials mentioned blinding of participants and personnel with the exception of one trial (*Kortam, Abdelrahman & Fateen, 2020*), which resulted (*El Sharkwy & Sharaf El-Din, 2019*; *El Sharkwy & Abd El Aziz, 2019*; *Ismail et al., 2014*; *Jamilian et al., 2017*; *Jamilian et al., 2019a*; *Jamilian et al., 2019b*; *Samimi et al., 2016*; *Talari et al., 2019*), in an unclear risk of bias. Seven trials had a low risk of bias (*El Sharkwy & Sharaf El-Din, 2019*; *El Sharkwy & Abd El Aziz, 2019*; *Ismail et al., 2014*; *Jamilian et al., 2017*; *Jamilian et al., 2019a*; *Jamilian et al., 2019b*; *Samimi et al., 2016*), highlighting that the patients and physicians were blinded to the treatment allocation. Only one trial (*Talari et al., 2019*) mentioned that the researchers and participants were not blinded to the allocation concealment, thereby resulting in a high risk of bias.
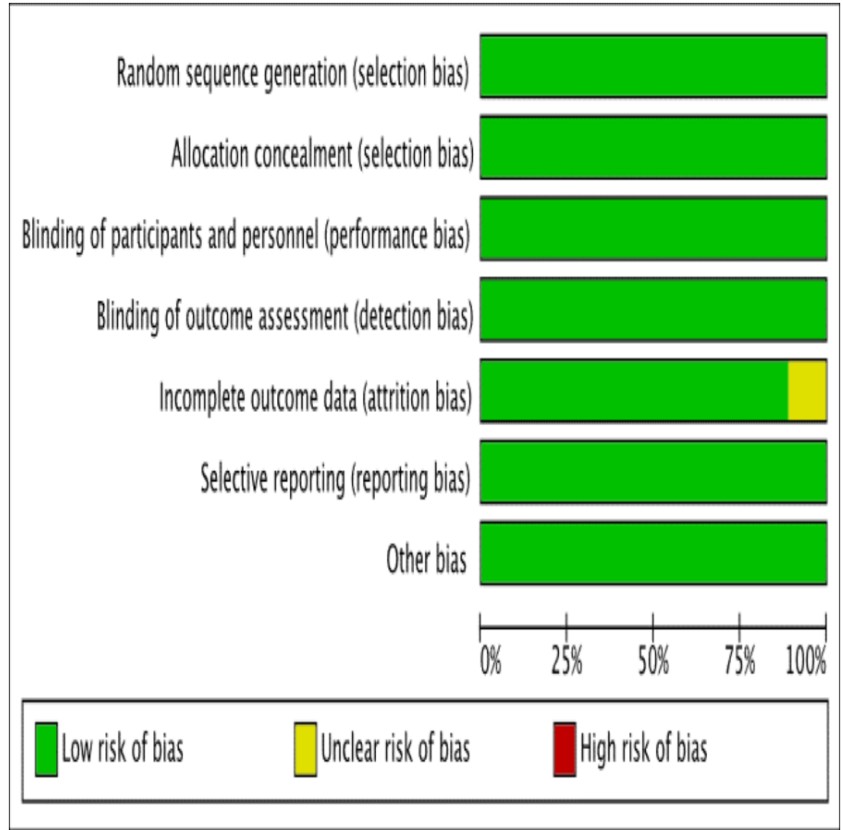

**Figure 2** **Risk of bias.** Risk of bias graph: authors' judgements about each risk of bias item presented as percentages across all included studies.

Eight trials reported the number of participants who completed the study, including the number of patients who dropped out from the study with justified reasons (*El Sharkwy & Sharaf El-Din, 2019*; *El Sharkwy & Abd El Aziz, 2019*; *Ismail et al., 2014*; *Jamilian et al., 2017*; *Jamilian et al., 2019a*; *Jamilian et al., 2019b*; *Samimi et al., 2016*; *Talari et al., 2019*). The missing participants for these trials were less than 15 percent (*El Sharkwy & Sharaf El-Din, 2019*; *El Sharkwy & Abd El Aziz, 2019*; *Ismail et al., 2014*; *Jamilian et al., 2017*; *Jamilian et al., 2019a*; *Jamilian et al., 2019b*; *Samimi et al., 2016*; *Talari et al., 2019*), and one trial (*Talari et al., 2019*) did not have any missing participants from both the control and intervention groups. Only one trial (*Kortam, Abdelrahman & Fateen, 2020*) did not mention the number of participants who completed or withdrew from the study. Neither did it summarize the patients' flow diagram, resulting in an unclear risk of bias.

All nine trials reported the outcomes as specified in their methods section (*El Sharkwy & Sharaf El-Din, 2019*; *El Sharkwy & Abd El Aziz, 2019*; *Ismail et al., 2014*; *Jamilian et al., 2017*; *Jamilian et al., 2019a*; *Jamilian et al., 2019b*; *Kortam, Abdelrahman & Fateen, 2020*; *Samimi et al., 2016*; *Talari et al., 2019*). Four trials registered their protocols, and three trials (*Jamilian et al., 2017*; *Jamilian et al., 2019a*; *Samimi et al., 2016*) were registered in

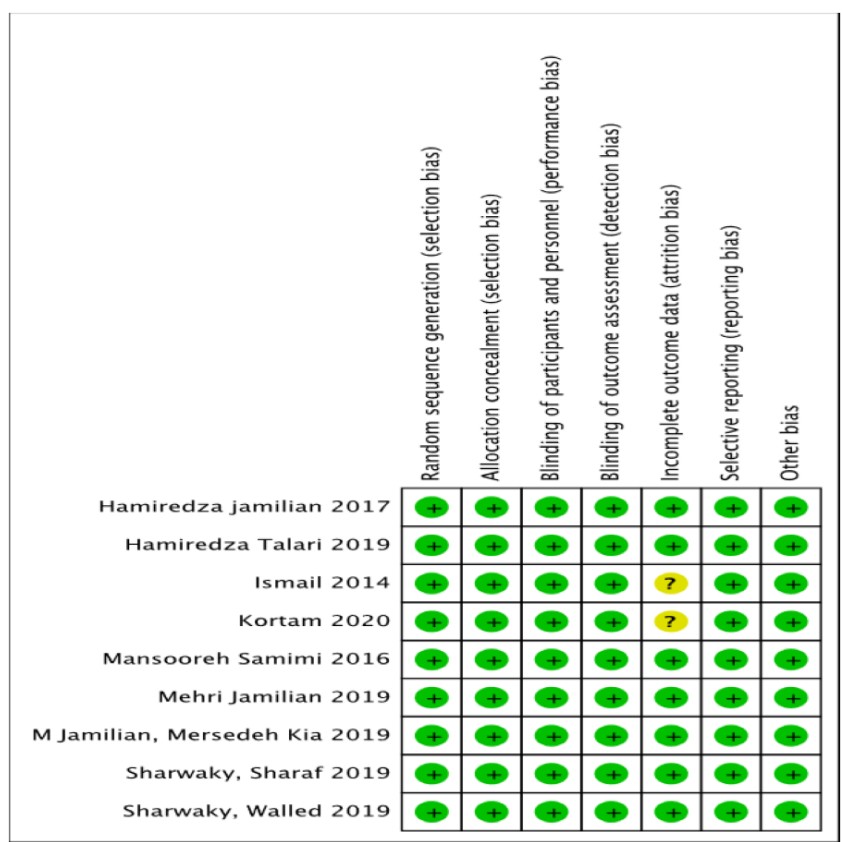

**Figure 3** **Risk of bias summary.** Risk of bias summary: authors' judgements on each risk of bias item for each included study.

the Iranian Registry of Clinical Trials. Only one trial (*El Sharkwy & Abd El Aziz, 2019*) was registered in the National Clinical Trials.

## LC supplementation for women with PCOS

We performed meta-analysis for each of the five comparison groups in this review. For the first comparison, the combination of clomiphene citrate and LC was compared with the combination of clomiphene citrate and placebo (*Ismail et al., 2014*; *Kortam, Abdelrahman & Fateen, 2020*). In total, 250 milligrams (mg) of oral clomiphene citrate was administered along with 3 grams (g) of LC in one study in comparison with the same 250 mg clomiphene citrate combined with placebo (*Ismail et al., 2014*). In another study, the researchers used 100 mg clomiphene citrate daily in combination with 3 g of LC when compared with the use of 100 mg clomiphene citrate and the placebo (*Kortam, Abdelrahman & Fateen, 2020*). The second comparison comprised the study that used 150 mg clomiphene citrate, 850 mg metformin and 1 g LC versus 150 mg clomiphene citrate, 850 mg metformin, and placebo (*El Sharkwy & Sharaf El-Din, 2019*). The third comparison included the studies that used a combination of 150 g clomiphene citrate and 600 mg oral N-Acetylcysteine in comparison with 150 mg clomiphene citrate and 3 g LC (*El Sharkwy & Abd El Aziz,*

**Table 1** Characteristic of the included studies.

| Studies | Participants | L carnitine dosage | Intervention | Comparison | Duration of intervention |
|---|---|---|---|---|---|
| (*El Sharkwy & Sharaf El-Din, 2019*) | Intervention, $n = 140$ Control, $n = 140$ | 3 g LC daily | 150 mg/day CC plus oral LC 3g and metformin 850 mg (1 tablet daily) | 150 mg/d CC plus metformin and placebo capsules | 12 weeks |
| (*El Sharkwy & Abd El Aziz, 2019*) | Intervention, $n = 82$ Control, $n = 82$ | 3 g LC daily | 150 mg/day of CC plus 3 g of oral LC daily, and placebo sachets | 150 mg/day of CC from day 3 until day 7 of the menstrual cycle plus 600 mg of oral $N$-acetylcysteine three times daily, and a placebo capsule | 12 weeks |
| (*Ismail et al., 2014*) | Intervention, $n = 85$ Control, $n = 85$ | 3 g LC daily | 250 mg CC from day three until day seven of the cycle plus LC 3 g daily | 250 mg CC with placebo | 12 weeks |
| (*Jamilian et al., 2017*) | Intervention, $n = 30$ Control, $n = 30$ | 250 mg LC | 250 mg carnitine supplements | Placebos (cellulose) | 12 weeks |
| (*Jamilian et al., 2019a*) | Intervention, $n = 26$ Control, $n = 27$ | 1,000 mg LC daily | LC 1,000 mg/d plus 200 mg/d chromium as chromium picolinate | Placebo | 12 weeks |
| (*Jamilian et al., 2019b*) | Intervention, $n = 27$ Control, $n = 27$ | 1,000 mg LC daily | 200 µg/day chromium picolinate plus 1,000 mg/day LC | Placebo (starch) | 12 weeks |
| (*Samimi et al., 2016*) | Intervention, $n = 30$ Control, $n = 30$ | 250 mg LC | 250 mg LC (capsule range 237–275 mg) | Placebo (cellulose) | 12 weeks |
| (*Talari et al., 2019*) | Intervention, $n = 30$ Control, $n = 30$ | 250 mg LC daily | 250 mg/day of LC | Placebo | 12 weeks |
| (*Kortam, Abdelrahman & Fateen, 2020*) | Intervention, $n = 47$ Control, $n = 47$ | 3g LC daily | Oral CC (50 mg tablet, two times per day) plus oral LC supplementation (1 g tablet, three times per day) | Oral CC only (50 mg tablet, two times per day). | Not stated |

*2019*). The fourth comparison included the studies that used 250 mg of LC in comparison with placebo (*Jamilian et al., 2017*; *Samimi et al., 2016*; *Talari et al., 2019*). Finally, the fifth comparison comprised the studies that used 200 mg chromium picolinate and 1g LC daily in comparison with the placebo (*Jamilian et al., 2019a*; *Jamilian et al., 2019b*).

## Comparison 1: Clomiphene citrate and LC versus clomiphene citrate and placebo

We performed meta-analysis in this comparison. No difference was observed in terms of the clinical pregnancy rate between the two groups (Risk ratio (RR) 7.12, 95% CI [0.14–350.06]; $I^2$ = 90%, $P$ = 0.32; two trials, $n$ = 264; low quality evidence) (*Ismail et al., 2014*; *Kortam, Abdelrahman & Fateen, 2020*). However, a difference was observed in terms of the primary outcome, ovulation rate between the two groups, which favored combination with placebo (RR 2.37, 95% CI [0.99–5.66]; $I^2$ = 88%, $P$ = 0.05; two trials, $n$ = 264; low quality evidence) (*Ismail et al., 2014*; *Kortam, Abdelrahman & Fateen, 2020*). Figure 4 showed the Forest plot, comparing the use of clomiphene citrate and LC in comparison with the use of clomiphene citrate and placebo for primary outcomes, clinical pregnancy rate, and ovulation rate. There is a difference in terms of the primary outcome, BMI within one group, which favored combination with placebo (MD 1.10, 95% CI [0.32–1.88]; $P$ = 0.006; one trial, $n$ = 94; moderate quality evidence) (*Kortam, Abdelrahman & Fateen, 2020*). No difference is observed for the secondary outcome, FSH within one group (MD −0.10, 95% CI, [−0.50–0.70]; $P$ = 0.75; one trial, $n$ = 94; moderate quality evidence) (*Kortam, Abdelrahman & Fateen, 2020*). There is no difference for the secondary outcome, LH within one group (MD (95% CI) −0.20 (−0.91–0.51); $P$ = 0.58; one trial, $n$ = 94; moderate quality evidence) (*Kortam, Abdelrahman & Fateen, 2020*). Therefore, in this comparison group, there was no significant difference in the pregnancy rate, FSH, and LH levels. However, there was a significant difference, favoring the placebo in terms of the ovulation rate and BMI. Table 2 showed the summary of findings and GRADE quality assessment for primary and secondary outcomes of Comparison 1.

## Comparison 2: Clomiphene citrate, metformin, and LC versus clomiphene citrate, metformin, and placebo

We performed meta-analysis in this comparison. There is a significant difference in the primary outcome, clinical pregnancy rate in one group, which favored combination with placebo (RR 4.27, 95% CI [2.15–8.47]; $P$ = 0.0001; one trial, $n$ = 274; moderate quality evidence) (*El Sharkwy & Sharaf El-Din, 2019*). There is a significant difference in the ovulation rate in one group, which favored combination with placebo (RR 3.15 95% CI [1.86–5.35]; $P$ = 0.0001; one trial, $n$ = 274; moderate quality evidence) (*El Sharkwy & Sharaf El-Din, 2019*). There is a significant difference for BMI in one group, which favored combination with placebo (MD 1.10, 95% CI [0.32–1.88]; $P$ = 0.006; one trial, $n$ = 274; moderate quality evidence) (*El Sharkwy & Sharaf El-Din, 2019*). There is a significant difference for the primary outcome, FPG in one group, which favored combination with LC (MD −5.10, 95% CI [−6.25 to −3.95]; $P$ = 0.00001; one trial, $n$ = 274; moderate quality evidence) (*El Sharkwy & Sharaf El-Din, 2019*) (Table 3). In addition, there is a

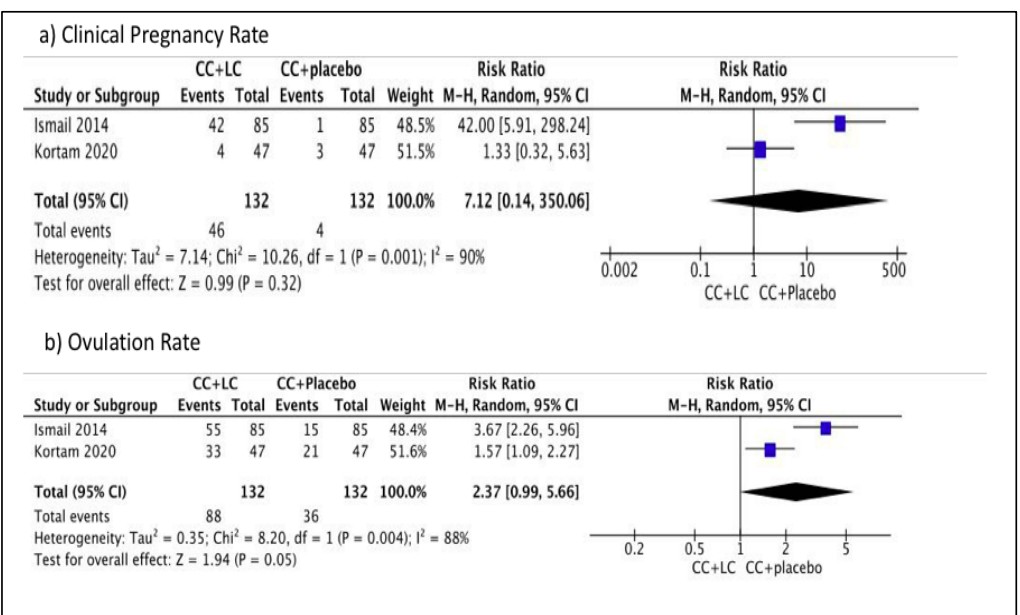

**Figure 4** Forest plot comparing clomiphene citrate and LC versus clomiphene citrate plus placebo for primary outcomes, clinical pregnancy rate and ovulation rate.

significant difference for the primary outcomes, LDL level in one group, which favored combination with LC (MD $-25.00$, 95% CI [$-27.93$ to $-22.07$]; $P = 0.00001$; one trial, $n = 274$; moderate quality evidence) (*El Sharkwy & Sharaf El-Din, 2019*), TC level in one group, which favored combination with LC (MD $-21.00$, 95% CI [$-24.14$ to $-17.86$]; $P = 0.00001$; one trial, $n = 274$; moderate quality evidence) (*El Sharkwy & Sharaf El-Din, 2019*) and TG level in one group, which favored combination with LC (MD $-9.00$, 95% CI [$-11.46$ to $-6.54$]; $P = 0.00001$; one trial, $n = 274$; moderate quality evidence) (*El Sharkwy & Sharaf El-Din, 2019*). There is a significant difference for the primary outcome, HDL level in one group, which favored combination with placebo (MD 15.50, 95% CI [12.42–18.58]; $P = 0.00001$; one trial, $n = 274$; moderate quality evidence) (*El Sharkwy & Sharaf El-Din, 2019*) (Table 3). There is a significant difference for the secondary outcomes, FSH level in one group, which favored combination with LC (MD $-0.63$, 95% CI [$-0.92$ to $-0.34$]; $P = 0.00001$; one trial, $n = 274$; moderate quality evidence) (*El Sharkwy & Sharaf El-Din, 2019*). There is a significant difference for LH level in one group, which favored combination with LC (MD$-2.36$, 95% CI [$-3.04$ to $-1.68$]; $P = 0.00001$; one trial, $n = 274$; moderate quality evidence) (*El Sharkwy & Sharaf El-Din, 2019*). In this comparison, there was a significant difference that favored combination with LC in terms of the FPG, LDL, TC, TG, HDL, FSH, and LH levels. There was a significant difference favoring the combination with placebo in pregnancy rate, ovulation rate, HDL level, and BMI. Table 3 showed the summary of finding and GRADE quality assessment for primary and secondary outcomes of Comparison 1.

Mohd Shukri et al. (2022), *PeerJ*, DOI 10.7717/peerj.13992

**Table 2  GRADE quality assessment for comparison 1: Comparing clomiphene citrate plus LC versus clomiphene citrate plus placebo.**

| | | Certainty assessment | | | | Number of patients | | Effect | | Certainty |
|---|---|---|---|---|---|---|---|---|---|---|
| Total study | Risk of bias | Inconsistency | Indirectness | Imprecision | Other considerations | LC + CC | CC + placebo | Relative (95% CI) | Absolute (95% CI) | |
| **Outcome: Clinical pregnancy rate** | | | | | | | | | | |
| 2 RCTs | not serious | serious[a] | not serious | serious[b] | none | 46/132 (34.8%) | 4/132 (3.0%) | RR 7.12 (0.14 to 350.06) | 185 more per 1,000 (from 26 fewer to 1,000 more) | ⊕⊕◯◯ LOW |
| **Outcome: Ovulation rate** | | | | | | | | | | |
| 2 RCTs | not serious | serious[a] | not serious | serious[b] | none | 88/132 (66.7%) | 36/132 (27.3%) | RR 2.37 (0.99 to 5.66) | 374 more per 1,000 (from 3 fewer to 1,000 more) | ⊕⊕◯◯ LOW |
| **Outcome: BMI** | | | | | | | | | | |
| 1 RCT | not serious | not serious | not serious | serious[c] | none | 47 | 47 | – | MD 0.4 lower (2.12 lower to 1.32 higher) | ⊕⊕⊕◯ MODERATE |
| **Outcome: Serum FSH** | | | | | | | | | | |
| 1 RCT | not serious | not serious | not serious | serious[c] | none | 47 | 47 | – | MD 0.1 higher (0.5 lower to 0.7 higher) | ⊕⊕⊕◯ MODERATE |
| **Outcome: Serum LH** | | | | | | | | | | |
| 1 RCT | not serious | not serious | not serious | serious[c] | none | 47 | 47 | – | MD 0.2 lower (0.91 lower to 0.51 higher) | ⊕⊕⊕◯ MODERATE |

**Notes.**

CI, Confidence interval; RR, Risk ratio; MD, Mean difference; RCT, Randomized controlled trial.

GRADE Working Group grades of evidence.

High certainty: We are very confident that the true effect lies close to that of the estimate of the effect.

Moderate certainty: We are moderately confident in the effect estimate: The true effect is likely to be close to the estimate of the effect, but there is a possibility that it is substantially different.

Low certainty: Our confidence in the effect estimate is limited: The true effect may be substantially different from the estimate of the effect.

Very low certainty: We have very little confidence in the effect estimate: The true effect is likely to be substantially different from the estimate of effect.

**Explanations**

[a] heterogeneity >75%.

[b] number of events<400.

[c] number of participants <400.

Mohd Shukri et al. (2022), *PeerJ*, DOI 10.7717/peerj.13992

**Table 3** GRADE quality assessment of Comparison 2: comparing clomiphene citrate, metformin plus LC versus clomiphene citrate, metformin plus placebo.

| | Certainty assessment | | | | | Number of patients | | Effect | | Certainty |
|---|---|---|---|---|---|---|---|---|---|---|
| Total study | Risk of bias | Inconsistency | Indirectness | Imprecision | Other considerations | LC +CC + MTF | CC + MTF + placebo | Relative (95% CI) | Absolute (95% CI) | |
| **Outcome: Clinical pregnancy rate** | | | | | | | | | | |
| 1 RCT | not serious | not serious | not serious | serious[a] | none | 39/138 (28.3%) | 9/136 (6.6%) | RR 4.27 (2.15 to 8.47) | 216 more per 1,000 (from 76 more to 494 more) | ⊕⊕⊕◯ MODERATE |
| **Outcome: Ovulation rate** | | | | | | | | | | |
| 1 RCT | not serious | not serious | not serious | serious[a] | none | 48/138 (34.8%) | 15/136 (11.0%) | RR 3.15 (1.86 to 5.35) | 237 more per 1,000 (from 95 more to 480 more) | ⊕⊕⊕◯ MODERATE |
| **Outcome: BMI** | | | | | | | | | | |
| 1 RCT | not serious | not serious | not serious | serious[b] | none | 138 | 136 | – | MD 1.1 higher (0.32 higher to 1.88 higher) | ⊕⊕⊕◯ MODERATE |
| **Outcome: Serum FPG** | | | | | | | | | | |
| 1 RCT | not serious | not serious | not serious | serious[b] | none | 138 | 136 | – | MD 5.1 lower (6.25 lower to 3.95 lower) | ⊕⊕⊕◯ MODERATE |
| **Outcome: Serum LDL** | | | | | | | | | | |
| 1 RCT | not serious | not serious | not serious | serious[b] | none | 138 | 136 | – | MD 25 lower (27.93 lower to 22.07 lower) | ⊕⊕⊕◯ MODERATE |
| **Outcome: Serum total cholesterol** | | | | | | | | | | |
| 1 RCT | not serious | not serious | not serious | serious[b] | none | 138 | 136 | – | MD 21 lower (24.14 lower to 17.86 lower) | ⊕⊕⊕◯ MODERATE |
| **Outcome: Serum HDL** | | | | | | | | | | |
| 1 RCT | not serious | not serious | not serious | serious[b] | none | 138 | 136 | – | MD 15.5 higher (12.42 higher to 18.58 higher) | ⊕⊕⊕◯ MODERATE |
| **Outcome: Serum triglyceride** | | | | | | | | | | |
| 1 RCT | not serious | not serious | not serious | serious[b] | none | 138 | 136 | – | MD 9 lower (11.46 lower to 6.54 lower) | ⊕⊕⊕◯ MODERATE |
| **Outcome: serum FSH** | | | | | | | | | | |
| 1 RCT | not serious | not serious | not serious | serious[b] | none | 138 | 136 | – | MD 0.63 lower (0.92 lower to 0.34 lower) | ⊕⊕⊕◯ MODERATE |
| **Outcome: serum LH** | | | | | | | | | | |
| 1 RCT | not serious | not serious | not serious | serious[b] | none | 138 | 136 | – | MD 2.36 lower (3.04 lower to 1.68 lower) | ⊕⊕⊕◯ MODERATE |

**Notes.**

CI, Confidence interval; RR, Risk ratio; MD, Mean difference; RCT, Randomized controlled trial; GRADE, Working Group grades of evidence (*GDT, 2022*).

High certainty: We are very confident that the true effect lies close to that of the estimate of the effect.

Moderate certainty: We are moderately confident in the effect estimate: The true effect is likely to be close to the estimate of the effect, but there is a possibility that it is substantially different.

Low certainty: Our confidence in the effect estimate is limited: The true effect may be substantially different from the estimate of the effect.

Very low certainty: We have very little confidence in the effect estimate: The true effect is likely to be substantially different from the estimate of effect.

**Explanations**

[a] number of events <400.

[b] number of participants <400.

## Comparison 3: Clomiphene citrate plus LC versus clomiphene citrate plus n-acetylcysteine

We performed meta-analysis in this comparison. There is no difference for the primary outcome, clinical pregnancy rate in one group (RR (95% CI) 1.16 (0.72, 1.89); $P = 0.54$; one trials, $n = 162$; moderate quality evidence) (*El Sharkwy & Abd El Aziz, 2019*). There is no difference for the primary outcome, ovulation rate in one group (RR (95% CI) 1.11 (0.79, 1.56); $P = 0.54$; one trials, $n = 162$; moderate quality evidence) (*El Sharkwy & Abd El Aziz, 2019*). There is no difference for the primary outcome, BMI in one group (MD 0.10, 95% CI [−0.78–0.98]; $P = 0.82$; one trial, $n = 162$; moderate quality evidence) (*El Sharkwy & Abd El Aziz, 2019*). There is a significant difference for the primary outcome, FPG in one group, which favored combination with NAC (MD 2.30, 95% CI [1.02–3.58]; $P = 0.0004$; one trial, $n = 162$; moderate quality evidence) (*El Sharkwy & Abd El Aziz, 2019*). There is a significant difference for the primary outcome, LDL level in one group, which favored combination with LC (MD −12.00, 95% CI [−15.80 to −8.20]; $P = 0.00001$; one trial, $n = 162$; moderate quality evidence) (*El Sharkwy & Abd El Aziz, 2019*). There is a significant difference for the primary outcome, TC level in one group, which favored combination with LC (MD −24.00, 95% CI [−27.61 to −20.39]; $P = 0.00001$; one trial, $n = 162$; moderate quality evidence) (*El Sharkwy & Abd El Aziz, 2019*). There is a significant difference for the primary outcome, HDL level in one group, which favored combination with NAC (MD 9.60, 95% CI [5.30–13.90]; $P = 0.0001$; one trial, $n = 162$; moderate quality evidence) (*El Sharkwy & Abd El Aziz, 2019*). There is a significant difference for the primary outcome, TG level in one group, which favored combination with LC (MD −19.00, 95% CI [−22.79 to −15.21]; $P = 0.00001$; one trial, $n = 162$; moderate quality evidence) (*El Sharkwy & Abd El Aziz, 2019*). The summary of all findings and GRADE quality assessment for primary outcomes of Comparison 3 is shown in Table 4.

There is a significant difference for the secondary outcome, FSH level in one group, which favored combination with LC (MD −0.50, 95% CI [−0.84 to −0.16]; $P = 0.004$; one trial, $n = 162$; moderate quality evidence) (*El Sharkwy & Abd El Aziz, 2019*). There is no difference for the secondary outcome, LH level in one group (MD −0.40, 95% CI [−1.51–0.71]; $P = 0.48$; one trial, $n = 162$; moderate quality evidence) (*El Sharkwy & Abd El Aziz, 2019*). In this comparison, there was no significant difference in the pregnancy rate, ovulation rate, BMI, and LH level. There was a significant difference that favored the combination of LC in LDL, TC, TG, and FSH levels, and there was a significant difference that favored the combination with NAC in terms of the FPG and HDL levels. The summary of all findings and GRADE quality assessment for secondary outcomes of Comparison 3 is shown in Table 4.

## Comparison 4: comparing LC with the placebo

We performed meta-analysis in this comparison. There was no difference for FPG in one group (MD −1.26, 95% CI [−7.50–4.98]); $P = 0.69$; one trial, $n = 60$; moderate quality evidence) (*Samimi et al., 2016*), LDL level in one group (MD 0.33, 95% CI [−0.05–0.71]; $P = 0.09$; one trial, $n = 60$; moderate quality evidence) (*Samimi et al., 2016*), total cholesterol level in one group (MD 6.84, 95% CI [−0.45–14.13]; $P = 0.07$; one trial, $n = 60$;

Mohd Shukri et al. (2022), *PeerJ*, DOI 10.7717/peerj.13992

**Table 4** Summary of findings and GRADE quality assessment of primary and secondary outcomes for Comparison 3: comparing clomiphene citrate plus LC versus clomiphene citrate plus n acetylcysteine.

| Total study | Risk of bias | Inconsistency | Indirectness | Imprecision | Other considerations | LC+CC | CC + NAC | Relative (95% CI) | Absolute (95% CI) | Certainty |
|---|---|---|---|---|---|---|---|---|---|---|
| | | Certainty assessment | | | | Number of patients | | Effect | | Certainty |
| **Outcome: Clinical pregnancy rate** | | | | | | | | | | |
| 1 RCT | not serious | not serious | not serious | serious[a] | none | 25/80 (31.3%) | 22/82 (26.8%) | RR 1.16 (0.72 to 1.89) | 43 more per 1,000 (from 75 fewer to 239 more) | ⨁⨁⨁◯ MODERATE |
| **Outcome: Ovulation rate** | | | | | | | | | | |
| 1 RCT | not serious | not serious | not serious | serious[a] | none | 38/80 (47.5%) | 35/82 (42.7%) | RR 1.11 (0.79 to 1.56) | 47 more per 1,000 (from 90 fewer to 239 more) | ⨁⨁⨁◯ MODERATE |
| **Outcome: BMI** | | | | | | | | | | |
| 1 RCT | not serious | not serious | not serious | serious[b] | none | 80 | 82 | – | MD 0.1 higher (0.78 lower to 0.98 higher) | ⨁⨁⨁◯ MODERATE |
| **Outcome: Serum FPG** | | | | | | | | | | |
| 1 RCT | not serious | not serious | not serious | serious[b] | none | 80 | 82 | – | MD 2.3 higher (1.02 higher to 3.58 higher) | ⨁⨁⨁◯ MODERATE |
| **Outcome: serum LDL** | | | | | | | | | | |
| 1 RCT | not serious | not serious | not serious | serious[b] | none | 80 | 82 | – | MD 12 lower (15.8 lower to 8.2 lower) | ⨁⨁⨁◯ MODERATE |
| **Outcome: serum total cholesterol** | | | | | | | | | | |
| 1 RCT | not serious | not serious | not serious | serious[b] | none | 80 | 82 | – | MD 24 lower (27.61 lower to 20.39 lower) | ⨁⨁⨁◯ MODERATE |
| **Outcome: serum HDL** | | | | | | | | | | |
| 1 RCT | not serious | not serious | not serious | serious[b] | none | 80 | 82 | – | MD 9.6 higher (5.3 higher to 13.9 higher) | ⨁⨁⨁◯ MODERATE |
| **Outcome: serum triglyceride** | | | | | | | | | | |
| 1 RCT | not serious | not serious | not serious | serious[b] | none | 80 | 82 | – | MD 19 lower (22.79 lower to 15.21 lower) | ⨁⨁⨁◯ MODERATE |
| **Outcome: serum FSH** | | | | | | | | | | |
| 1 RCT | not serious | not serious | not serious | serious[b] | none | 80 | 82 | – | MD 0.5 lower (0.84 lower to 0.16 lower) | ⨁⨁⨁◯ MODERATE |
| **Outcome: serum LH** | | | | | | | | | | |
| 1 RCT | not serious | not serious | not serious | serious[b] | none | 80 | 82 | – | MD 0.4 lower (1.51 lower to 0.71 higher) | ⨁⨁⨁◯ MODERATE |

**Notes.**

CI, Confidence interval; RR, Risk ratio; MD, Mean difference; RCT, Randomized controlled trial; GRADE, Working Group grades of evidence.

High certainty: We are very confident that the true effect lies close to that of the estimate of the effect.

Moderate certainty: We are moderately confident in the effect estimate: The true effect is likely to be close to the estimate of the effect, but there is a possibility that it is substantially different.

Low certainty: Our confidence in the effect estimate is limited: The true effect may be substantially different from the estimate of the effect.

Very low certainty: We have very little confidence in the effect estimate: The true effect is likely to be substantially different from the estimate of effect.

**Explanations**

[a] number of events <400.

[b] number of participants <400.

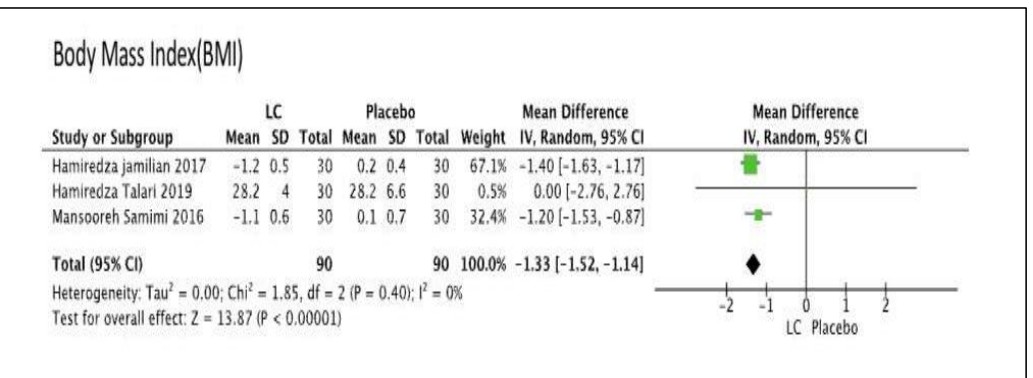

**Figure 5** Forest plot for the primary outcome, body mass index (BMI) of comparison 4: comparing of the LC versus the placebo.

moderate quality evidence) (*Samimi et al., 2016*), HDL level in one group (MD 0.00, 95% CI [−3.60–3.60]; $P = 1.00$; one trial, $n = 60$; moderate quality evidence) (*Samimi et al., 2016*), and TG level in one group (MD 0.15, 95% CI [−0.14–0.44]; $P = 1.00$; one trial, $n = 60$; moderate quality evidence) (*Samimi et al., 2016*). There was significant difference for BMI level in three groups, which favored LC group (MD −1.33, 95% CI [−1.52 to −1.44]; $I^2 = 0\%$, $P = 0.00001$; three trials, $n = 180$; moderate quality evidence) (*Jamilian et al., 2017*; *Samimi et al., 2016*; *Talari et al., 2019*). Figure 5 showed the Forest plot of Comparison 4, comparing LC with placebo for the primary outcome, BMI. The summary of findings of primary outcomes and GRADE quality assessment for Comparison 4 is shown in Table 5.

There is a significant difference for the secondary outcome, mental health status, by using assessment score, BDI score in one group, which favored placebo (MD 2.50, 95% CI [2.35–2.65]; $P = 0.00001$; one trial, $n = 60$; moderate quality evidence) (*Jamilian et al., 2017*), general health questionnaire (GHQ) score in one group, which favored LC (MD −5.80, 95% CI [−6.10 to −5.50]; $P = 0.00001$; one trial, $n = 60$; moderate quality evidence) (*Jamilian et al., 2017*), and depression anxiety stress score (DASS) in one group, which favored LC (MD −6.80, 95% CI [−7.20 to −6.40]; $P = 0.00001$; one trials, $n = 60$; moderate quality evidence) (*Jamilian et al., 2017*). Therefore, in this comparison, there was no significant difference in terms of the FPG, LDL, TC, HDL, and TG levels, whereas there were significant differences that favored LC with respect to BMI, GHQ, and DASS scores, and significant differences that favored placebo in the BDI score. The summary of findings of secondary outcomes and GRADE quality assessment for Comparison 4 is shown in Table 5.

## Comparisons 5: LC plus chromium and placebo

We performed meta-analysis in this comparison. There is no difference for the primary outcome, FPG in one group (MD −3.40, 95% CI [−7.60–0.80]; $P = 0.11$; one trial, $n = 54$; moderate quality evidence) (*Jamilian et al., 2019b*). There is no difference for the primary outcome, LDL level in one group (MD −0.60, 95% CI [−19.95–18.75]; $P = 0.95$; one trial,

**Table 5  The summary of findings of outcomes and GRADE quality assessment for comparison 4: comparing of the LC versus the placebo.**

| Total study | Certainty assessment | | | | | Number of patients | | Effect | | Certainty |
|---|---|---|---|---|---|---|---|---|---|---|
| | Risk of bias | Inconsistency | Indirectness | Imprecision | Other considerations | LC | Placebo | Relative (95% CI) | Absolute (95% CI) | |
| **Outcome: Serum FPG** | | | | | | | | | | |
| 1 RCT | not serious | not serious | not serious | serious[a] | none | 30 | 30 | – | MD 1.26 lower (7.5 lower to 4.98 higher) | ⊕⊕⊕◯ MODERATE |
| **Outcome: Serum LDL** | | | | | | | | | | |
| 1 RCT | not serious | not serious | not serious | serious[a] | none | 30 | 30 | – | MD 0.33 higher (0.05 lower to 0.71 higher) | ⊕⊕⊕◯ MODERATE |
| **Outcome: Serum total cholesterol** | | | | | | | | | | |
| 1 RCT | not serious | not serious | not serious | serious[a] | none | 30 | 30 | – | MD 6.84 higher (0.45 lower to 14.13 higher) | ⊕⊕⊕◯ MODERATE |
| **Outcome: Serum HDL** | | | | | | | | | | |
| 1 RCT | not serious | not serious | not serious | serious[a] | none | 30 | 30 | – | MD 0 (3.6 lower to 3.6 higher) | ⊕⊕⊕◯ MODERATE |
| **Outcome: Serum Triglyceride** | | | | | | | | | | |
| 1 RCT | not serious | not serious | not serious | serious[a] | none | 30 | 30 | – | MD 0.15 higher (0.14 lower to 0.44 higher) | ⊕⊕⊕◯ MODERATE |
| **Outcome: Serum BMI** | | | | | | | | | | |
| 3 RCTs | not serious | not serious | not serious | serious[a] | none | 90 | 90 | – | MD 1.33 lower (1.52 lower to 1.14 lower) | ⊕⊕⊕◯ MODERATE |
| **Outcome: Mental health status (using BDI)** | | | | | | | | | | |
| 1 RCT | not serious | not serious | not serious | serious[a] | none | 30 | 30 | – | MD 2.5 higher (2.35 higher to 2.65 higher) | ⊕⊕⊕◯ MODERATE |
| **Outcome: Mental health status (using GHQ)** | | | | | | | | | | |
| 1 RCT | not serious | not serious | not serious | serious[a] | none | 30 | 30 | – | MD 5.8 lower (6.1 lower to 5.5 lower) | ⊕⊕⊕◯ MODERATE |
| **Outcome: Mental health status (using DASS)** | | | | | | | | | | |
| 1 RCT | not serious | not serious | not serious | serious[a] | none | 30 | 30 | – | MD 6.8 lower (7.2 lower to 6.4 lower) | ⊕⊕⊕◯ MODERATE |

**Notes.**

CI, Confidence interval; MD, Mean difference; RCT, Randomized controlled trial; BDI, Beck Depression Index; GHQ, General Health Questionnaire; DASS, Depression Anxiety Stress Score; GRADE, Working Group grades of evidence.

High certainty: We are very confident that the true effect lies close to that of the estimate of the effect.

Moderate certainty: We are moderately confident in the effect estimate: The true effect is likely to be close to the estimate of the effect, but there is a possibility that it is substantially different.

Low certainty: Our confidence in the effect estimate is limited: The true effect may be substantially different from the estimate of the effect.

Very low certainty: We have very little confidence in the effect estimate: The true effect is likely to be substantially different from the estimate of effect.

**Explanations**.

[a] number of participants <400.

$n = 54$; moderate quality evidence) (*Jamilian et al., 2019b*). There is no difference for the primary outcome, TC in one group (MD $-9.70$, 95% CI [$-28.53$–$9.13$]; $P = 0.31$; one trial, $n = 54$; moderate quality evidence) (*Jamilian et al., 2019b*). There is no difference for the primary outcome, HDL level in one group (MD $-3.40$, 95% CI [$-8.20$–$1.40$]; $P = 0.17$; one trial, $n = 54$ moderate quality evidence) (*Jamilian et al., 2019b*). There is significance difference for the primary outcome, TG level in one group, which favored combination with LC (MD $-28.10$, 95% CI [$-47.25$ to $-8.95$]; $P = 0.004$; one trial, $n = 54$; moderate quality evidence) (*Jamilian et al., 2019b*). The summary of primary outcomes' findings and GRADE quality assessment is shown in Table 6.

There is no difference for the secondary outcome, mental health status, by using BDI scoring in one group (MD $-1.50$, 95% CI [$-4.17$–$1.17$]; $P = 0.27$; one trial, $n = 53$; moderate quality evidence) (*Jamilian et al., 2019a*), GHQ scoring in one group (MD $-1.80$, 95% CI [$-7.10$–$3.50$]; $P = 0.51$; one trial, $n = 53$; moderate quality evidence) (*Jamilian et al., 2019a*), and DASS scoring in one group (MD $-3.50$, 95% CI [$-11.42$–$4.42$]; $P = 0.39$; one trial, $n = 53$; moderate quality evidence) (*Jamilian et al., 2019a*). Therefore, in this comparison, there was no difference in FPG, LDL, TC, HDL, BDI score, GHQ score, and DASS score. On the other hand, there was a significant difference that favored combination with LC in terms of the TG level. The summary of secondary outcomes' findings and GRADE quality assessment is shown in Table 6.

## DISCUSSION

Menstrual problems, hyperandrogenism, and infertility are the most common symptoms observed during the early reproductive years in PCOS (*Peigné & Dewailly, 2014*). Pregnancy-specific complications, obesity, glucose intolerance, type 2 diabetes, cardiovascular diseases, and gynecological malignancies can all develop as women get older. For these at-risk women, lifelong monitoring is required, and preventative actions need to be implemented early (*Peigné & Dewailly, 2014*). The health risks associated with PCOS may extend far beyond the management of the common presenting symptoms or fertility treatment, as this disease and its symptoms are likely to last beyond the reproductive age until menopause (*Cooney & Dokras, 2018*). The scope of studies has been limited in terms of evaluating the risk for cardiovascular morbidity and mortality in women with PCOS after they undergo menopause.

This review was designed to include all RCTs addressing the effect of LC supplementation in women with PCOS. The nine selected trials had created a diverse group, addressing various comparisons and outcomes, thereby resulting in several comparisons that contributed to each of our predefined outcomes. We were unable to perform subgroup analyses, as there were inadequate trials that used similar comparisons.

To evaluate the impact of LC on PCOS patients, we conducted a comprehensive literature study. From nine trials, only five trials can be sub-grouped into similar combination of comparisons, wherein two trials (*Ismail et al., 2014*; *Kortam, Abdelrahman & Fateen, 2020*) in Comparison 1 were associated with the outcomes of clinical pregnancy rate and ovulation rate, and three trials (*Jamilian et al., 2017*; *Samimi et al., 2016*; *Talari et al., 2019*)

Peer**J**

**Table 6  The summary of primary and secondary outcome findings and GRADE quality assessments for Comparison 5: comparing of LC plus chromium with the placebo.**

| Total study | Risk of bias | Inconsistency | Indirectness | Imprecision | Other considerations | LC + Chromium | placebo | Relative (95% CI) | Absolute (95% CI) | Certainty |
|---|---|---|---|---|---|---|---|---|---|---|
| | | Certainty assessment | | | | Number of patients | | Effect | | Certainty |
| **Outcome: Serum FPG** | | | | | | | | | | |
| 1 RCT | not serious | not serious | not serious | serious[a] | none | 27 | 27 | – | MD 3.4 lower (7.6 lower to 0.8 higher) | ⊕⊕⊕◯ MODERATE |
| **Outcome: Serum LDL** | | | | | | | | | | |
| 1 RCT | not serious | not serious | not serious | serious[a] | none | 27 | 27 | – | MD 0.6 lower (19.95 lower to 18.75 higher) | ⊕⊕⊕◯ MODERATE |
| **Outcome: Serum Total cholesterol** | | | | | | | | | | |
| 1 RCT | not serious | not serious | not serious | serious[a] | none | 27 | 27 | – | MD 9.7 lower (28.53 lower to 9.13 higher) | ⊕⊕⊕◯ MODERATE |
| **Outcome: Serum HDL** | | | | | | | | | | |
| 1 RCT | not serious | not serious | not serious | serious[a] | none | 27 | 27 | – | MD 3.4 lower (8.2 lower to 1.4 higher) | ⊕⊕⊕◯ MODERATE |
| **Outcome: Serum Triglyceride** | | | | | | | | | | |
| 1 RCT | not serious | not serious | not serious | serious[a] | none | 27 | 27 | – | MD 28.1 lower (47.25 lower to 8.95 lower) | ⊕⊕⊕◯ MODERATE |
| **Outcome: Mental health status (using BDI)** | | | | | | | | | | |
| 1 RCT | not serious | not serious | not serious | serious[a] | none | 26 | 27 | – | MD 1.5 lower (4.17 lower to 1.17 higher) | ⊕⊕⊕◯ MODERATE |
| **Outcome: Mental health status (using GHQ)** | | | | | | | | | | |
| 1 RCT | not serious | not serious | not serious | serious[a] | none | 26 | 27 | – | MD 1.8 lower (7.1 lower to 3.5 higher) | ⊕⊕⊕◯ MODERATE |
| **Outcome: Mental health status (using DASS)** | | | | | | | | | | |
| 1 RCT | not serious | not serious | not serious | serious[a] | none | 26 | 27 | – | MD 3.5 lower (11.42 lower to 4.42 higher) | ⊕⊕⊕◯ MODERATE |

**Notes.**

CI, Confidence interval; MD, Mean difference; RCT, Randomized controlled trial; BDI, Beck Depression Index; GHQ, General Health Questionnaire; DASS, Depression Anxiety Stress Score; GRADE, Working Group grades of evidence.

High certainty: We are very confident that the true effect lies close to that of the estimate of the effect.

Moderate certainty: We are moderately confident in the effect estimate: The true effect is likely to be close to the estimate of the effect, but there is a possibility that it is substantially different.

Low certainty: Our confidence in the effect estimate is limited: The true effect may be substantially different from the estimate of the effect.

Very low certainty: We have very little confidence in the effect estimate: The true effect is likely to be substantially different from the estimate of effect.

**Explanations**.

[a] number of participants <400.

in Comparison 4 were related to BMI outcomes. Thus, as a result, the application of the findings in this review is limited. On the outcome basis, three primary outcomes, namely clinical pregnancy rate, ovulation rate, and FPG, have similar trials with similar combination of comparisons, in which two trials were related to clinical pregnancy rates, two trials were associated with ovulation rate, and three trials were focused on FPG. From the reported incidence of adverse events, we detected side effects in one trial (*Kortam, Abdelrahman & Fateen, 2020*), that is, abdominal pain, dizziness, and nausea. However, none of the trial investigators reported serious side effects due to the use of LC. Most of PCOS women have issues with infertility. Given the scarcity of trials comparing similar comparisons, future clinical trials comparing LC alone with other comparators in similar comparisons are needed to determine the effect of LC on improving pregnancy rate and ovulation rate in PCOS patients. The overall quality of the evidence used in this review ranges from moderate to low. The trials differed in terms of comparison type and supplementation dosage. We also recommend that future trials consider using standardized LC dosages, regimes, and consumption durations, either alone or in combination, to produce homogeneous results across trials to demonstrate the safety and effectiveness of the LC.

The overall quality of the evidence contributing to this review ranges from moderate to low. The type of comparison and supplementation dosage varied among the trials. Most trials had low risk of bias for allocation bias with the exception of one trial (*Jamilian et al., 2019b*), as randomization was manually performed at the clinic. In terms of the blinding of participants and personnel, one trial (*Kortam, Abdelrahman & Fateen, 2020*) had unclear risk of bias, and one trial (*Talari et al., 2019*) had high risk of bias, as the researchers and participants were not blinded in their trial. All trials had reported outcomes in their method section, whereas four trials published their protocols. The risk of attrition bias was only observed in one trial (*Kortam, Abdelrahman & Fateen, 2020*), as it did not state the number of participants who withdrew from the study or completed the study. The percentage of participants who failed to follow-up was less than 15 percent in eight trials, and two trials (*Jamilian et al., 2017*; *Talari et al., 2019*) declared that financing had been received from the university grant. We encountered high heterogeneity in the meta-analysis, and we were unable to segment any further because there were insufficient trials in each group comparison. Even though all of the included studies showed the same direction of effect, we found significant heterogeneity in our primary outcomes. Due to the small number of trials, we were unable to conduct subgroup analysis.

We aimed to reduce the publication bias by searching different databases without language restrictions and examining the reference lists of all linked articles for additional references. Unfortunately, we cannot guarantee that we have discovered all the trials in this area. As only nine trials were included, we could not create a funnel plot to detect bias or heterogeneity, and not all included trials reported similar outcomes. Although all the included studies showed the same direction of effect, we encountered high heterogeneity in our primary outcomes. We could not perform sub-group analysis due to limited number of trials.

One systematic review has examined the impact of LC on patients with PCOS (*Maleki et al., 2019*). The researchers in this review evaluated the potential roles played by LC in PCOS

patients. It included two observational studies (*Celik et al., 2017*; *Fenkci et al., 2008*) and four randomized controlled studies, wherein three studies (*Ismail et al., 2014*; *Jamilian et al., 2019b*; *Samimi et al., 2016*) were included in this meta-analysis, and one study (*Latifian, Hamdi & Totakhneh, 2015*) was unrelated to our primary and secondary outcomes. Similar to our meta-analysis, the BMI had a significant impact on LC supplementation based on three trials (*Ismail et al., 2014*; *Jamilian et al., 2019b*; *Samimi et al., 2016*). However, for the lipid profile, one study had a significant impact (*Ismail et al., 2014*), whereas two studies had an insignificant impact (*Fenkci et al., 2008*; *Samimi et al., 2016*).

## CONCLUSIONS

Based on this meta-analysis, it has been observed that LC is beneficial for improving BMI as well as LDL, TC, and TG levels, in women with PCOS. However, in terms of the clinical pregnancy rate and ovulation rate, the meta-analysis showed insignificant effect. Therefore, the justification of LC usage for these outcomes requires further evaluations and clinical trials. The findings of this review would need to be considered in the context of LC, as supplementation with other medications in the treatment of PCOS. In this study, the scope of evaluation of the side effects of LC use is limited, and more safety data is needed to assess the risks of using it. If further studies are conducted to examine the use of LC in women with PCOS, they should include pregnancy rate and ovulation rate as part of their outcomes. This is because PCOS women mostly seek treatment to alleviate fertility problems. Data on physical findings such as hirsutism, acne, and weight reduction can also be considered in the subsequent research studies.

### Registration and Protocol

Our systematic review and meta-analysis protocol has been registered and published in PROSPERO (Registration number: CRD42021232433).

### Funding

The authors received no funding for this work.

### Competing Interests

Norhayati Mohd Noor is an Academic Editor for PeerJ.

### Author Contributions

- Mohd Falihin Mohd Shukri conceived and designed the experiments, performed the experiments, analyzed the data, prepared figures and/or tables, authored or reviewed drafts of the article, and approved the final draft.
- Mohd Noor Norhayati conceived and designed the experiments, authored or reviewed drafts of the article, and approved the final draft.
- Salziyan Badrin conceived and designed the experiments, performed the experiments, analyzed the data, prepared figures and/or tables, authored or reviewed drafts of the article, and approved the final draft.

- Azidah Abdul Kadir conceived and designed the experiments, performed the experiments, analyzed the data, authored or reviewed drafts of the article, and approved the final draft.

## Data Availability

The PROSPERO protocol, forest plot, search strategy, PRISMA checklist are available in the Supplementary File.

## Supplemental Information

Supplemental information for this article can be found online at http://dx.doi.org/10.7717/peerj.13992#supplemental-information.

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
