# Peer review of "Effects of L-carnitine supplementation for women with polycystic ovary syndrome: a systematic review and meta-analysis"

_PeerJ, doi:10.7717/peerj.13992_

## Round 0.1 · original submission · Minor Revisions

Thank you for the submission. In reviewing the information provided by the reviewers, I am requesting revisions before the article can be considered further. Please address reviewer 1 and reviewer 2 comments in your revision.

Reviewer 1 ·

Basic reporting

The article is clearly written

Experimental design

The author reported use of Cochrane Central Register of Controlled Trials (CENTRAL), MEDLINE,
Embase, Cumulative Index to Nursing and Allied Health Literature (CINAHL), Psychological
Information Database (PsycINFO), and the World Health Organization International Clinical
Trials Registry Platform for retrieval of artcles. However, while reporting the PRISMA flow diagram its not possible to retrieve only 52 studies.
The authors need to revise the search stratergy to correctly identify the relevant articles for the current hypothesis of systematic review and metaanalysis.

Validity of the findings

The full search stratergy need to be provided in greater detail. The article does not add anything new to the literature.

Reviewer 2 ·

Basic reporting

Please refer to the additional comments

Experimental design

Please refer to the additional comments

Validity of the findings

Please refer to the additional comments

Additional comments

I read this article with interest. This systematic review and meta-analysis assessed the beneficial effects of L carnitine on BMI, lipid profile, and pregnancy outcomes among women with polycystic ovarian syndrome (PCOS). This topic is helpful in the field of reproductive endocrinology and infertility. However, the lack of enough studies in this field limits the conclusion drawn by the authors in this paper. Nevertheless, the authors did a good job in summarizing the available evidence from the literature.
I have some suggestions and comments to the authors to improve their manuscript. The authors are advised to check typographical and grammatical errors in the manuscript. I have listed some of them below but there are still several more errors that need to be corrected.

Abstract
Line 7 Add “is” before “characterized”
Line 8 Add a comma before which and change “leading” to “leads”
Line 13: It was not clear in the methods what were the primary outcomes of the study.
Line 23 Change “random-effect model” to “random-effects model”

Introduction
Line 76-77 The objective was not clear. What do you want to determine? Effectiveness of LC in temporizing symptoms of PCOS? Treating the obesity and dyslipidemia? Treating hyperandrogenism? Improve fertility? Please make it clear.

Methods
Line 120-121: Please change “were not fulfilling’ to “did not fulfill”
Line 141 Please add “of” before “risk of bias”
Line 143 How many authors assessed the risk of bias?
Line 148 Change “authors” to “author”

Results
For each comparison group, please mention whether you performed meta-analysis or you were just discussing the qualitative synthesis of the included studies.
For the results for Comparison 2, please make a short synthesis of the study. It’s difficult to understand the main message that the authors want to convey from this comparison group.
Line 191 Change “Characteristic” to “Characteristics”
Line 197 Please add “were” in between “who” and “diagnosed”.
Line 204 Please add “were” in between “who” and “pregnant”
Line 302 Change “significance” to “significant”
Line 304 Change “significance” to “significant”

Discussion
Line 421-423 Can the authors check if this statement is correct: “However, there were significant differences in primary outcome, clinical pregnancy rate and ovulation rate which favoured combination with the placebo.”. When I looked at the forest plot, I did not see any significance in the pooled RR. Even the p-values are greater than or equal to 0.05.
Please discuss the limitations of this systematic review and how future clinical trials can be improved.

Conclusion
The authors use L-carnitine and L carnitine interchangeably. Please decide which one to use and make it uniform in the whole manuscript.
Line 469-471 “BMI has significant improvement in comparison with L carnitine versus placebo, and in contrast, in lipid profile particularly LDL, TC and TG level had a significant effect on PCOS patients.” Was this statement based on meta-analysis or based on individual studies? Please clarify.

Reviewer 3 ·

Basic reporting

The article is written in English, clear and easy to understand the point of view of authors. There are some grammar mistakes which can be corrected by proof read. References quoted clearly. Figures and tables are clear easy to navigate and understand. Raw data shared is helpful.

Experimental design

Aims and scope of this study is clear. Methods used are clear and described in details.

Validity of the findings

The review impact is moderate. This is because there are limited studies which focus exactly on the effects of LC and its underlying mechanism in PCOS pathophysiology. most studies focusing on improving symptoms encountered by patients with PCOS. All underlying data for the metanalysis is provided and is statistically sound. Conclusion is comprehensive and well written.

Annotated reviews are not available for download in order to protect the identity of reviewers who chose to remain anonymous.

---

## Round 0.2 · Minor Revisions

Thank you very much for resubmitting the modified version of the manuscript. However, there are a few comments that need to be revised before the manuscript can be accepted.

1) Although the authors have mentioned that they have corrected the grammatical and typographical errors, there are still more errors that need to be corrected in this manuscript. I suggest sending the manuscript to a professional proofreader. Besides that, please check the abbreviation used for L-carnitine throughout the manuscript.

3) Please refrain from repeating your findings in the discussion section. Instead, the discussion should emphasise what is new, different, or important about your results and consider the alternative explanations.

3) Please discuss the limitations of this systematic review and how future clinical trials can be improved. 

4) Please add the revisions that have been made in the author's response file rather than just mentioning that "we have revised" or "we have made corrections".

Reviewer 2 ·

Basic reporting

The authors have addressed all my comments and questions.

Experimental design

The authors have addressed all my comments and questions.

Validity of the findings

The authors have addressed all my comments and questions.

Additional comments

The authors have addressed all my comments and questions.

---

## Round 0.3 · Minor Revisions

Dear Authors,

Thank you for resubmitting the revised version of this manuscript.

May I have the revised version that has been proofread by the copyediting service (PeerJ), as you mentioned in the rebuttal's letter?

Please update the term Polycystic Ovarian Syndrome to Polycystic Ovary Syndrome.

Thank you very much.

---

## Round 0.4 · accepted · Accept

Thank you for resubmitting the revised version of this manuscript. The manuscript has been significantly improved, and I recommend the acceptance of this manuscript.